# Do Pharmacological Treatments Act in Collaboration with Rehabilitation in Spinal Cord Injury Treatment? A Review of Preclinical Studies

**DOI:** 10.3390/cells13050412

**Published:** 2024-02-27

**Authors:** Syoichi Tashiro, Shinsuke Shibata, Narihito Nagoshi, Liang Zhang, Shin Yamada, Tetsuya Tsuji, Masaya Nakamura, Hideyuki Okano

**Affiliations:** 1Department of Rehabilitation Medicine, School of Medicine, Keio University, Tokyo 160-8582, Japan; 2Department of Rehabilitation Medicine, Faculty of Medicine, Kyorin University, Tokyo 181-8611, Japan; 3Division of Microscopic Anatomy, Graduate School of Medical and Dental Sciences, Niigata University, Niigata 951-8510, Japan; 4Department of Orthopaedic Surgery, School of Medicine, Keio University, Tokyo 160-8582, Japan; 5Department of Physiology, School of Medicine, Keio University, Tokyo 160-8582, Japan

**Keywords:** plasticity, neuromodulation, neuroprotection, regeneration, neurotrophic factor, scar, exercise, physiotherapy, physical therapy, paresis

## Abstract

There is no choice other than rehabilitation as a practical medical treatment to restore impairments or improve activities after acute treatment in people with spinal cord injury (SCI); however, the effect is unremarkable. Therefore, researchers have been seeking effective pharmacological treatments. These will, hopefully, exert a greater effect when combined with rehabilitation. However, no review has specifically summarized the combinatorial effects of rehabilitation with various medical agents. In the current review, which included 43 articles, we summarized the combinatorial effects according to the properties of the medical agents, namely neuromodulation, neurotrophic factors, counteraction to inhibitory factors, and others. The recovery processes promoted by rehabilitation include the regeneration of tracts, neuroprotection, scar tissue reorganization, plasticity of spinal circuits, microenvironmental change in the spinal cord, and enforcement of the musculoskeletal system, which are additive, complementary, or even synergistic with medication in many cases. However, there are some cases that lack interaction or even demonstrate competition between medication and rehabilitation. A large fraction of the combinatorial mechanisms remains to be elucidated, and very few studies have investigated complex combinations of these agents or targeted chronically injured spinal cords.

## 1. Introduction

Spinal cord injury (SCI) is a severe condition that induces permanent disabilities including paresis, sensory disturbance, spasticity, and bowel and rectal disorder. Functional recovery of SCI is observed typically within the first 6 months after injury in the acute-to-subacute phase. In the chronic phase, no further functional improvement is usually expected [1,2]. Rehabilitation may be the most popular and feasible and least invasive and costly treatment without associated ethical concerns for improving or maintaining the remaining function of patients after the subacute phase of SCI. Rehabilitative training includes various methods such as functional training of paretic limbs and muscles, physical exercise, and reconditioning. In addition, advanced rehabilitation measures are being developed, including body-weight-supported (BWS) treadmill training (TMT) [3], robot-assisted gait training [4,5], functional electrical stimulation [6], and virtual reality training [7]. Many preclinical studies have shown that molecular and histological changes mainly underlie motor functional recovery [8]. Representative histological changes include regenerative and plastic changes in the intraspinal circuit [9,10,11,12,13] and descending spinal tracts [14], the enforcement of synaptic connectivity [15], the restoration of the spinal inhibitory capacity to improve motor control and coordination or to suppress spasticity, and sensory–motor integration [16]. Expression of molecules is reportedly modified secondary to training, with upregulation of various neurotrophic factors, including brain-derived neurotrophic factor (BDNF), neurotrophin 3 (NT3), neurotrophin 4 (NT4), glial-cell-line-derived neurotrophic factor (GDNF) [17], nerve growth factor (NGF) [18], and insulin-like growth factor-1 (IGF-1) in acute-to-subacute SCI [19]. Neurotrophic factors promote neural plasticity, vascularization, and neuroprotection and are considered to underlie the abovementioned beneficial changes [20]. Furthermore, it was reported that physical-activity-mediated functional recovery involves endogenous neural stem cells (NSCs) [21] and that TMT promotes the proliferation and migration of ependymal cells, an endogenous source of NSCs [22]. Another critical aspect of rehabilitative training is amelioration of the negative impact of disuse secondary to SCI [23]. Training can prevent the decline in muscle volume and function and the change in muscular fiber type [24]. While it is often overlooked in preclinical studies of chronic SCI, disuse-induced functional deterioration is speculated to suppress or even mask the beneficial effects of specific treatments [25]. While rehabilitation promotes body functionality and activity of daily living in the acute-to-subacute patients, the treatment effects are insufficient to induce remarkable functional recovery in the chronic phase. The extraordinary efforts of clinicians and researchers are currently being devoted to establishing more effective training methods and to clarifying the mechanisms underlying rehabilitation targeting chronic SCI.

Combinatorial treatment to enhance the effects of rehabilitation is another prominent theme of preclinical studies in SCI research. While rehabilitation induces various beneficial changes, as represented by the promotion of neuronal plasticity and regeneration and modification of microenvironments in the injured spinal cord on its own, and these changes are directly linked to functional recovery, additional treatments may support and enhance the effect. Pharmacological treatments, including medication, biological agents such as neurotrophic factors and biomaterials, cell transplantation, and physical medicine modalities, will be good candidates because they represent the most common therapeutic strategy for diseases and have advantages in terms of their non-invasiveness, feasibility, and ethicality. However, to our knowledge, there are no reviews regarding combinatorial therapy with a variety of medical agents and rehabilitation. Although this research field has a relatively long history, relatively few studies have fairly evaluated the effects of combinatorial treatment. Some early studies seem to have an insufficient design, such as a lack of appropriate controls, heterogeneity of intervention conditions, a lack of combinations, and an insufficient sample size. This fact has made it difficult for researchers to distinguish which combinatorial effect might be significant in reality. In addition, the difference of treatment effects, which are rooted in the strategy of rehabilitative training, is sometimes overlooked by non-clinician researchers, even though some combinations have shown competing effects. On these grounds, we first summarized combinatorial treatments with rehabilitation and medication. In addition, we focused also on the cases in which these two modalities showed competition. This review will help facilitate preclinical and clinical research on rehabilitation and medication and further develop SCI treatments.

## 2. Materials and Methods

We searched the *Web of Science* (BIOSIS), *Medline* (via PubMed), and *Scopus* databases for studies published by 30 September 2023. Keyword combinations of “spinal cord injury” and “rehabilitation” or related words were applied. The following keywords were applied to identify studies about rehabilitation: “exercise”, “training”, or “task-specific”. Studies applying medication and biological agents including neurotrophic factors were included, while studies applying cell and gene therapies, viral induction, scaffold single treatment, and physical medicine modalities were excluded. However, we included studies in which a certain treatment was used in combination with rehabilitative training and physical medicine treatment because physical medicine treatment is frequently coupled with or even included in rehabilitation and studies applying a scaffold loaded with a medical agent. On the other hand, we excluded studies in which cell therapy or nerve transplantation was applied as the third factor (i.e., training + medication + cells), including genetically modified cell transplantation.

Preclinical research was screened based on its title and abstract. The current review only included studies that compared the effects of combinatorial treatment with those of single rehabilitative treatments because it is essential to determine whether combinatorial treatments have synergistic effects compared with single rehabilitative treatments. In addition, studies comparing the states before and after medication in the same animals were excluded because the lasting effects of previous medication and the effects of sequential medications are usually not assessed. Of the 104 studies that were selected and checked throughout, 43 were included in the tables. Combinatorial treatments were classified into four domains: neuromodulation, neurotrophic factors, agents that counteract inhibitory factors among glial and fibrotic scar components, and others. Since the topic and main result of each study vary, we chose to accumulate short introductions of each single study to be as precise as possible in this review.

## 3. Results

### 3.1. Neuromodulation

Most investigators used a serotonergic pharmacotherapy strategy in this treatment area. We detected twelve studies in this category and briefly summarized the basic information, the content of rehabilitation and medical treatments, and the chief mechanisms (Table 1). Quipazine, a nonspecific 5-HT agonist that acts on the 5HT2A and 5HT1 receptors and improves motor performance in spinal animals, is the most commonly used agent to modify spinal neurological activity [26,27]. It is often used in combination with another serotonergic agent called 8-OH-DPAT, a 5-HT1A/7 receptor agonist, to stimulate the serotonergic system more broadly. In addition, epidural electrical stimulation (EES) is sometimes used as a physiologic means to enhance the treatment effect.

#### 3.1.1. Quipazine and 8-OH-DPAT

De Leon et al. combined quipazine and a passive robotic bipedal treadmill early in 2006, but this showed no synergistic effect [28]. Courtine et al. used dual-site EES at the L2 and S1 levels as well as medication with quipazine and 8-OH-DPAT in combination with bipedal TMT. They showed that dual-site EES and either medication immediately enabled a full-weight-bearing gait in spinal-cord-transected rats and was useful for rehabilitation. While motor recovery was promoted by each single treatment in combination with TMT, the full combinatorial treatment with TMT restored the motor-evoked potential (MEP), reduced c-Fos expression, and improved kinematics and electromyography patterns [29]. Courtine’s group further developed a novel robotically supported bipedal overground gait-training apparatus and established electrochemical neuromodulation treatments composed of medication with quipazine, 8-OH-DPAT, and a dopamine D1 receptor agonist; dual-site EES; and multisystem rehabilitation to apply different rehabilitative strategies according to the grade of motor recovery. They showed that remodeling of cortical projections involving neuronal relays at various levels enabled qualitative control of electrochemically stimulated lumbosacral circuitries to execute refined locomotion according to contextual information. They further found that multisystem rehabilitation had a markedly better effect than automated single TMT [30]. While they mostly used the same interventions as this previous study, Asboth et al. reported the generalization effect on natural locomotor function of the treatment set [14]. Recently, this group further combined deep brain stimulation and reported synergistic facilitation of locomotion [31]. From the field of physical medicine and rehabilitation, the application of neuromodulation and brain–computer interface to integrate neuromechanical and pharmacological interventions was proposed [32]. Another research group revealed a different aspect of a similar treatment strategy. Ganzer et al. reported that combinatorial treatment comprising passive bicycle training and serotonergic pharmacotherapy with quipazine and 8-OH-DPAT induced dose-dependent cortical reorganization extended to the unaffected forelimb [33], increased weight-supported stepping, normalized 5-HT receptor density, and reversed dendritic atrophy [34]. Yao et al. used wireless EES during locomotor training in combination with quipazine and 8-OH-DPAT. They found that combinatorial treatment with EES improved basal metabolism and that the higher the EES frequency, the better the gait appearance, with higher task metabolism [35]. On the other hand, Foffani et al. investigated the difference in the combinatorial effect according to the training method: active walking and passive cycling. They found that quipazine enhanced cortical reorganization induced by TMT but limited that induced by cycling [27]. Thus, quipazine acts by competing in some of the training modalities. While the mechanism remained to be elucidated, the authors concluded that single treatment with quipazine but not quipazine + 8-OH-DPAT might interfere with the upregulation of BDNF and/or adenylate cyclase 1 induced by passive bike exercise [36].

#### 3.1.2. Commercially Available Serotonin Agonists

Quipazine is sometimes substituted by commercially available serotonin agonists such as fluoxetine, a selective serotonin reuptake inhibitor, and buspirone, an anti-anxiety 5-HT_1A_ receptor agonist. Cristante et al. treated acute SCI model rats with TMT in combination with fluoxetine and found more remarkable functional improvement accompanied by a better MEP [37]. Ryu et al. compared the effects of fluoxetine and cyproheptadine, a 5-HT_2_ receptor antagonist, in late-subacute SCI model rats. While spastic behavior and electrophysiological abnormalities were suppressed by TMT and cyproheptadine treatment, they were increased by fluoxetine treatment even in combination with TMT. They observed no significant difference in locomotor function or the density of synaptophysin-positive puncta among groups, including nontreated control animals [38]. This study suggests that spasticity should not be overlooked as a potential adverse effect of serotonergic pharmacotherapy. The reason why Ryu et al. did not detect significant locomotor recovery may be related to the phase of SCI; chronic SCI is refractory to various treatments, and the characteristics of the late-subacute phase are similar to those of chronic SCI [20]. Ung et al. further applied buspirone, carbidopa, and L-DOPA in combination with TMT to activate the central pattern generator. They found that significant locomotor recovery and increased muscle volume were induced secondary to combinatorial treatment in the subacute phase. However, no additional beneficial effect was observed when they further combined clenbuterol, a β2-adrenergic agonist similar to anabolic steroids [39]. In a similar intervention, Liu et al. applied carbidopa, a decarboxylase inhibitor, and L-DOPA, i.e., a noradrenergic/dopaminergic precursor, together with passive bike exercise. They showed that exercise, medication, and combinatorial treatment each had comparable effects in suppressing spasticity in an electrophysiological evaluation [40].

**Table 1 cells-13-00412-t001:** Neuromodulatory Treatments Targeting the Lumbar Spinal Cord Applied with Rehabilitation.

Basic Information/Model	Rehabilitation	Combinatorial Treatment/Groups	Effect
2006 de Leon [28] SD rats (female, *N* = 28) T8/9 transection	Subacute started at 3 weeks -Robot-assisted training, 20 min/day, 5 days/week for 10 weeks	Subacute; started at 3 weeks -Quipazine: daily (i.t.), 2–3 min before training Groups: Combined, Trained, Quipazine	Only behavioral assessments. (1) More stepping movements in trained group than untrained; (2) No significant effect of combinatorial treatment on stepping recovery.
2016 Foffani G [27] SD rats (*N* = 70) T8/9 transection	Subacute start at 1 week -Quadrupedal TMT, 3 min/day, 5 days/week for 8 weeks -Bike training, two sessions of 30 min/day, 3 days/week for 8 weeks	Subacute; treated at 2 weeks -Quipazine: daily (i.p.) Groups: Bike, TMT, Quipazine, Bike + Quipazine, TMT + Quipazine, Bike + TMT + Quipazine	Responding cell number in the primary somatosensory cortex decreased in quipazine-treated animals in combination with bike exercise but increased with TMT. TMT and quipazine were collaborative, while bike and quipazine were competing.
2013 Ganzer PD [33] SD rats (female, *N* = 24) T8/9 transection	Subacute start at 1 week -BWS passive bicycling, two sessions of 30 min/day, 3 days/week for 8 weeks	Subacute; treated at 2 weeks -Quipazine, 8-OH-DPAT, 5 days/week for 8 weeks Groups: Saline + Rehab, Low dose + Rehab, High dose + Rehab	Combinatorial effect of training was not evaluated. (1) Dose-dependent reorganization of sensorimotor cortex, extended to cortex corresponding to unaffected forelimb.
2018 Ganzer PD [34] SD rats (female, *N* = 26; second experiment) T8/9 transection	Subacute start at 1 week -BWS passive bicycling, two sessions of 30 min/day, 3 days/week for 8 weeks	Subacute; treated at 2 weeks -Quipazine, 8-OH-DPAT, 5 days/week for 8 weeks Groups: Bike, Quipazine/8-OH-DPAT, Combined	(1) While TMT alone increased 5-HT receptor immunoreactivity, quipazine reversed it. Addition of TMT to quipazine did not exert any effect; (2) Addition of TMT restored MAP2-positive dendritic processes and improved weight-supported stepping.
2009 Courtine G [29] SD rats (female, *N* = 7–10, per group) T7 transection	Subacute start at 8 days -Partial BWS-TMT, 20 min/day, every other day for 8 weeks	Subacute; together with training -EES to L2 + S1 at 40–50 Hz, 1–4 V -Quipazine (i.p.)/8-OH-DPAT (s.c.) 10–15 min before training Groups: TMT + EES (L2 + S1), TMT + Quipazine + 8-OH-DPAT, Combined	Addition of BWSTT restored MEPs, reduced c-Fos expression, and improved kinematics and electromyography patterns.
2012 van den Brand R [30] SD rats (female, *N* = 27) T6 transection	Subacute start at 8 days -TMT alone or TMT + bipedal overground training with a robotic device, 30 min/day for 8 weeks	Subacute; together with training -EES to L2 + S1 at 40 Hz -Systemic injection of a cocktail of quipazine/8-OH-DPAT/ D1 agonist Groups: Drug, Drug + TMT, Drug + Overground/TMT	(1) CST fiber density was restored accompanying MEP amplitude restoration in overground gait-trained animals; (2) Upregulation in c-Fos expression secondary to overground gait training; (3) Automated treadmill-restricted training failed to promote trans-lesional plasticity and recovery.
2018 Asboth L [14] Lewis rats (female, *N* = 15) T8/9 severe contusion	Subacute start at 7 days -Bipedal TMT + overground training with a robotic device, 40 min/day, 6 days/week for 9 weeks	Subacute; together with training -EES to L2 + S1 at 40 Hz, 100–300 μA -Quipazine/8-OH-DPAT/D1 agonist 5 min before training Groups: Training, Training + EES, Training + EES + Drug	This study included a further investigation using mice without rehabilitation. (1) Combinatorial treatment improved parameters related to natural locomotion, stair climbing, and swimming; (2) Combination of training increased projections from cortex and ventral gigantocellular reticular neurons to lumbar enlargement.
2021 Yao Q [35] SD rats (female, *N* = 60) T8 transection	Subacute start at 1 week -High-intensity BWS bipedal gait, 30 min/day for 11 weeks	Subacute; together with training -EES (20/40/60 Hz) at T12–L2 during training -Quipazine (i.p.) and 8-OH-DPAT (s.c.) 10 min before training Groups: Training, Training + EES, Training + Quipazine/8-OH-DPAT, Combined	Combinatorial effect of training was not evaluated except for behavior. (1) EES restored gait rhythm; (2) 5-HT agonists increased task metabolism, which is essential to facilitate locomotor activity, in a frequency-dependent manner; (3) The sensitive stimulation frequency differed between segments: 40 Hz for L1–L2 and 60 Hz for T12–T13.
2010 Liu H [40] SD rats (female, *N* = 56) T8/9 transection	Subacutely started at 7 DPI -Passive motorized bicycle exercise, two 30 min/day sessions, 5 days/week for 30 days	Subacute, together with training Carbidopa (a decarboxylase inhibitor) + L-DOPA (a noradrenergic/dopaminergic precursor (p.o.) 30 min before training Groups: Training, Dopaminergic treatment, Combined	Electrophysiological measurement of spasticity improved in all treatment groups, thus no additive effect of training was observed.
2012 Ung RV [39] CD1 mice (male, *N* = 43) T9/10 transection	Subacute start at 1 week -Quadrupedal BWS-TMT, 15 min/day, 3 days/week for 8 weeks	Subacute; together with training (1) BCD: buspirone + carbidopa + L-DOPA (i.p.) coupled with training (3 days/week) (2) Clenbuterol (a β2-adrenergic agonist similar to anabolic steroids) (s.c.) daily Groups: TMT, TMT + BCD, Combined	Combinatorial effect of training was not evaluated; all groups had accompanying rehabilitative training. BCD + training induced locomotor recovery accompanied by larger muscle volume; clenbuterol increased muscle volume.
2013 Cristante AF [37] Wistar rats (male, *N* = 96) T9/10 moderate contusion	Acute start after SCI -TMT 15 min/day, 5 days/week for 6 weeks	Acute; together with training -Fluoxetine (a selective serotonin reuptake inhibitor) (i.p.) until 42 DPI Groups: TMT, Fluoxetine, Combined	Though each single treatment induced greater functional improvement accompanied by better MEP parameters, no combinatorial effect was observed. No effect on the residual fiber count.
2018 Ryu Y [38] SD rats (*N* = 64) T8 severe contusion	Late-subacute start at 4 weeks -BWS passive bipedal gait followed by quadrupedal gait, 30 min/day for 2 weeks	Late-subacute; together with training -Fluoxetine (i.p.) or cyproheptadine (a 5-HT_2_ receptor antagonist) Groups: TMT, Fluoxetine, Cyproheptadine, TMT + Fluoxetine, TMT + Cyproheptadine	(1) While spasticity was suppressed by cyproheptadine treatment and increased by fluoxetine, TMT did not modify those effects; (2) Addition of TMT effectively induced downregulation of 5-HT_2A_ receptor in combination groups as well as control.

BWS—body-weight-supported; DPI—days postinjury; EES—epidural electrical stimulation; MEP—motor-evoked potential; SCI—spinal cord injury; SD—Sprague–Dawley; TMT—treadmill training.

### 3.2. Neurotrophic Factors

Neurotrophic factors are essential for the survival and function of neural cells and are often reported to induce various therapeutic effects on SCI, including neural sparing, axonal regeneration and terminal arborization, and synaptogenesis [41]. These factors include “neurotrophic factors” in the narrow sense and other trophic factors that have similar vital effects on the neural system. The former include NGF, BDNF, NT3, and NT4, and the latter include GDNF, IGF-1, fibroblast growth factor 2 (FGF2), epidermal growth factor (EGF), vascular endothelial growth factor-A, platelet-derived growth factor-AA (PDGF-AA), pleiotrophin, and hepatocyte growth factor [41]. While researchers have demonstrated the therapeutic potential of these factors as monotherapies in preclinical and clinical studies of SCI, it is well recognized that rehabilitative training and neurotrophic factors elicit synergistic effects [42]. Interestingly, rehabilitative training upregulates some endogenous neurotrophic factors including NGF, BDNF, NT3, GDNF, and IGF-1 [17,18,19]. Thus, neurotrophic factors may be part of the mechanisms underlying the beneficial effects of rehabilitation and may enhance the effects of rehabilitation. Despite the wide variety of neurotrophic factors, only a few types (represented by BDNF and NT3) have been studied in the context of combinatorial treatment with rehabilitative training. We thus chose six studies that investigated the combinatorial effect of neurotrophic factors and rehabilitation (Table 2).

#### 3.2.1. BDNF

BDNF acts to promote axonal growth and sprouting as well as neuroprotective effects [41]. Marchionne et al. used bipedal TMT in combination with BDNF via intrathecal delivery using an osmotic pump throughout the experimental period in cats with acute spinal cord transection and reported a qualitative improvement [43]. Han et al. transplanted a collagen-binding, domain-tagged, BDNF-containing, linear-ordered collagen scaffold (LOCS) or blank LOCS, which may guide nerve regeneration, in a canine lower thoracic cord transection model and performed rehabilitation for all animals. They found that the BDNF-containing LOCS induced locomotor recovery, with restoration of somatosensory-evoked potentials, reduction of the lesion volume, recovery of neurofilaments and 5-HT-positive fibers, and remyelination [44].

#### 3.2.2. NT3

NT3 is one of the myokines expressed by muscle spindles and acts on the central nervous system after it traverses the blood–brain barrier [45]. The effect of NT3 on the injured spinal cord is mostly similar to BDNF. However there is a discussion regarding the nociceptive system. While NT3 ameliorates pain-related behavior, some researchers have reported that BDNF sometimes induces sensitization, particularly in an overexpression model with adeno-associated virus (AAV) [46]. Lin et al. applied NT3 with a biofunctional scaffold to thoracic-cord-transected rats and investigated the effects of combinatorial rehabilitation. They showed that TMT with a customized device that enables replication of normal hindlimb movements promoted motor recovery. Further investigation revealed that rehabilitation increased the level of anti-inflammatory cells and promoted axonal ingrowth into scaffolds and perineuronal net formation, which may lead to better locomotor recovery [47]. Tom et al. transplanted a bioengineered scaffold loaded with BDNF and NT3 [48]. While motor function did not significantly differ among groups, spasticity was improved, as assessed by the rate depression property of the H-reflex, and KCC2 expression was upregulated in the trained groups. In particular, KCC2 expression was further upregulated after combinatorial treatment. While it has been reported that training ameliorates spasticity via upregulation of KCC2 [49], the study by Tom at al. suggests that the addition of NT3 may strengthen the effect of training. Moreover, a research revealed that the KCC2 agonist restored stepping ability via the integration of descending inputs into the relay circuit [50]. One noteworthy study combined NT3 via AAV, spinal electromagnetic stimulation (EMS) and exercise intervention, and showed the addition of NT3 promoted recovery of locomotor function, the electrophysiological response, and retrograde tracer transport [51].

#### 3.2.3. Other Neurotrophic Factors

Acute treatment with granulocyte colony-stimulating factor [52] together with rehabilitative training enhanced locomotor recovery, in line with neuroprotective findings. A study showed the acute treatment effect of GDNF using an AAV expression model [53]. Alluin et al. used growth factor cocktails (GFs) composed of EGF, FGF2, PDGF-AA, and cABC in combination with training. While the individual effects of each agent were not investigated, the researchers found that the combinatorial pharmacotherapy induced synergistic effects in terms of neuroanatomical plasticity in collateral sprouting of CST and serotonergic fibers but without significant motor recovery [54].

**Table 2 cells-13-00412-t002:** Neurotrophic Factor Treatments Applied with Rehabilitation.

Basic Information/Model	Rehabilitation	Combinatorial Treatment	Effect
2020 Marchionne F [43] Domestic shorthair cats (adult, female, *N* = 14) T11/12 transection	Acute -Bipedal TMT, 20 min/day: forelimbs were kept stationary on the platform	Immediate -BDNF (i.t.) with an osmotic pump throughout the experimental period Groups: TMT, BDNF + TMT	Combinatorial effect of training was not evaluated; all groups had accompanying rehabilitative training. (1) BDNF-treated cats recovered weight-bearing plantar stepping, while control animals dragged H/L at a higher velocity qualitatively.
2015 Han S [44] Beagle dogs (adult, female, *N* = 28) T12 transection	Subacute start at 3 weeks -Daily massage and stretch, pinched foot, and suspended body for involuntary H/L movement -Ambulation with body support by holding the tail after wound healing	Immediate -LOCS fibers with/without collagen-binding BDNF implant Groups: LOCS + Training, LOCS-BDNF + Training	Combinatorial effect of training was not evaluated; all groups had accompanying rehabilitative training. LOCS-BDNF promoted locomotion and sensory recovery, accompanied by reductions of lesion volume and collagen deposition and increases of axonal regeneration and myelination.
2019 Lin J [47] SD rats (female, *N* = 26) T9/10 transection with a 2 mm gap	Subacute start at 1 week -Quadrupedal BWS-TMT with a device to enable normal gait, 30 min/day, 3 days/week for 11 weeks	Immediate implantation -Biofunctional scaffold: poly(ε-caprolactone-co-ethyl ethylene phosphate) loaded with NT3 Groups: Scaffold, Scaffold + TMT	Rehabilitated animals showed better motor function together with more regenerated axons, a higher percentage of anti-inflammatory M2-like macrophages, and greater perineuronal net formation than non-rehabilitated animals.
2018 Tom B [48] Rats (*N* = 56) T9/10 moderate contusion	Acute -BWS-TMT, 1000 steps/day, 5 day/week for 8 weeks	Immediate implantation -Bioengineered scaffold made of poly N-isopropylacrylamide-g-poly ethylene glycol loaded with BDNF/NT3 Groups: TMT, Scaffold-BDNF/NT3, Combined	(1) Motor function did not significantly differ among the groups; (2) Spasticity was improved, as assessed by the rate depression property of the H-reflex, in the trained groups; (3) KCC2 expression was restored in the trained groups.
2014 Alluin O [54] * Wistar rats (female, *N* = 48), Mid thoracic clip compression	Acute start on day 1 -Quadrupedal TMT, some of 15 min sessions/day, 5 days/week for 7 weeks	Acute; started at 4 DPI -cABC and growth factors (GFs: EGF, FGF2, and PDGF-AA), osmotic pump infusion, for 7 days Groups: TMT, cABC + GFs, Combined	(1) Absence of synergistic effects on kinematic parameters; (2) cABC effect to suppress astrogliosis was increased; (3) Synergistic effects on neuroanatomical plasticity in collateral sprouting of CST and serotonergic fibers.
2020 Park CH [52] SD rats (*N* = 24) T9 clip moderate contusion	Subacute start at 1 week -Quadrupedal gait exercise, 30 min/day, 5 days/week for 4 weeks	Acute intervention -G-CSF (i.p.) for 5 days Groups: Exercise, G-CSF, Combined	The G-CSF/exercise group showed the most effective functional recovery, the smallest cavity size, higher BDNF, and lower GFAP immunoreactivity. There was lower VEGF in the combination group than the single treatment groups, but it was still higher than in the control group.

BDNF—brain-derived neurotrophic factor; BWS—body-weight-supported; cABC—chondroitinase ABC; CST—corticospinal tract; DPI—days postinjury; EGF—epidermal growth factor; EMS—electromagnetic stimulation; FGF2—fibroblast growth factor 2; G-CSF—granulocyte colony-stimulating factor; GDNF—glial-cell-line-derived neurotrophic factor; H/L—hindlimb; KCC2—kalium-chloride cotransporter 2; LOCS—linear-ordered collagen scaffold; NT3—neurotrophin 3; PDGF-AA—platelet-derived growth factor; RtST—reticulospinal tract; SCI—spinal cord injury; SD—Sprague–Dawley; TMT—treadmill training; VEGF—vascular endothelial growth factor. *: a study also included in another table.

### 3.3. Agents Targeting Inhibitory Factors in Scar Tissue

In the chronically injured spinal cord, glial and fibrotic scars hinder the recovery process. A persistent low-grade inflammation after subacute SCI induces both fibrotic scar remodeling by fibroblast-like cells or macrophages to express inhibitory factors to plastic changes and glial scar replacement with components such as chondroitin sulfate proteoglycans (CSPGs), which form a barrier-like structure to obstruct axonal regeneration. Degradation of such inhibitory factors is another important treatment strategy [20]. Researchers have investigated the combinatory effect of these agents with rehabilitative interventions (Table 3). We found eight studies applying cABC to degrade CSPG, two of which used another agent targeting keratan sulfate degradation and anti-Nogo-A antibody, respectively, as a comparison. One study used a growth factor cocktail in combination with cABC as introduced above [54]. In addition, a few studies investigated unique agents.

#### 3.3.1. cABC

cABC is one of the most widely used agents to treat SCI, and it degrades CSPGs, a major component of glial scars [55]. Tester and Howland performed 1-month cABC treatment in combination with rehabilitation training consisting of bipedal TMT, overground walking, horizontal ladder crossing, narrow beam crossing, and a pegboard for food rewards in acute-phase SCI model cats. They found that combinatorial treatment with cABC enhanced recovery of skilled locomotion along with serotonergic plasticity [56]. Garcia-Alias et al. compared the treatment effect of specific and non-specific rehabilitation targeting forelimb function in combination with cABC treatment. They found cABC enhanced neural plasticity via enhancing CST sprouting. While a combination of cABC and specific rehabilitation improved manual dexterity of the forelimb, animals that underwent non-specific rehabilitation showed better ladder walking and worse skilled reaching abilities. Thus, they concluded that successful reinforcement of one behavior may interfere with another behavior [57]. Similarly, Wang et al. investigated the effect of combinatorial treatment from the late-subacute phase of SCI (4 weeks after injury) in rats. They also found that cABC treatment significantly improved recovery of skilled paw reaching and ladder and beam walking, and both cABC treatment and rehabilitation increased the CST branching and crossing of the gray/white matter boundary and BDA-labeled axons. Furthermore, rehabilitation synergistically increased the number of CST axons at the lesion epicenter and vesicular glutamate transporter 1 expression (VGLUT1)-positive presynaptic boutons. In addition, they found that the component of perineuronal nets was upregulated secondary to rehabilitation [58]. Prager et al. investigated the combinatorial effects of cABC and OECs, which promote axonal regeneration and remyelination, together with forepaw-reaching rehabilitation. They showed that transplantation of cABC-expressing OECs immediately after SCI increased CST sprouting into gray matter and the level of 5-HT-positive fibers caudal to the lesion, with consecutive modest functional improvement. While not included in the current review, this study is important because it shows the clear synergistic potential of medication in combination with cell therapy and rehabilitation [59]. On the other hand, Shinozaki et al. reported that combinatorial treatment of the chronically injured spinal cord with cABC only induced a small benefit in locomotor recovery but restored the transverse residual tissue area and induced neuronal/serotonergic fiber regeneration [60]. A recent study showed a beneficial effect of multimodal treatment with cortical epidural stimulation, 3 h a day, and forelimb behavioral rehabilitation in a C7 injury model in cABC treatment via lentiviral application [61]. Notably, combinatorial treatment with non-viral messenger RNA cABC delivery and rehabilitation have also been reported [62]. Finally, Alluin et al. applied cABC together with GFs as described above [54].

#### 3.3.2. Other Agents to Degrade CSPG

A research group induced disintegrin and metalloproteinase with thrombospondin motifs-4 (ADAMTS4) via AAV to catalyze the proteolysis of CSPG protein cores, showing a treatment effect [63]. Ishikawa et al. focused on another component of glial scars. They compared the combinatorial effects of cABC and keratanase II, which degrades keratan sulfate instead of CSPGs, with single-pellet reaching rehabilitation and found that the effects of keratanase II and cABC were comparable in terms of functional recovery and neurite growth [64].

#### 3.3.3. Anti-Nogo-A Treatments

Nogo-A is a myelin-associated neurite growth inhibitor. Blockade of Nogo-A induces axonal regeneration and functional recovery in SCI animals [65]. Maier et al. first performed bipedal and quadrupedal TMT in combination with anti-Nogo-A antibody treatment in subacute SCI animals, but this treatment did not elicit synergistic effects. Both treatments induced recovery, and anti-Nogo-A antibody treatment increased regeneration and neuronal reorganization as expected. They speculated that the lack of synergistic effects is due to an increase in pain perception or interference in the recovery process [66]. However, a study by Chen et al. using a similar rehabilitation method reported that combinatorial therapy improved motor function, with CST sprouting caudal to the lesion, increased serotonergic and VGLUT1-positive excitatory synapses to lumbar motoneurons, and the greatest reduction in lumbar interneuron activity as assessed by c-Fos expression. They also reported that the treatment had no effects on thermal nociception, mechanical allodynia, or lesion volume [67]. In addition, one study showed the combinatorial effects of training and Nogo66 receptor gene deletion [68]. These results suggest that Nogo-A blockade and rehabilitative training elicit effects via different mechanisms and do not always have synergistic effects.

Zhao et al. studied a clinically relevant strategy: a combination of an anti-Nogo-A antibody, cABC, and scar dissection together with forelimb and hindlimb rehabilitation consisting of Montoya-type staircase reaching and ladder walking. When used in combination with rehabilitation, they reported that each individual agent induced a similar degree of functional recovery, sprouting, and axonal regeneration, while the combination of the anti-Nogo-A antibody and cABC elicited more significant effects. Furthermore, they found that the anti-Nogo-A antibody stimulated growth of thicker axons, while cABC affected thinner axons [69].

#### 3.3.4. Inhibitors for Axonal Regrowth Inhibitor

Zhang et al. applied bipedal TMT with a semaphorin 3A inhibitor, which induces axonal regeneration and functional recovery by blocking the axonal growth inhibitor semaphorin 3A [70], through a unique drug delivery system: a silicon sheet like an artificial dura mater. They found that combinatorial treatment resulted in better gait kinematics than single treatment and induced the highest level of synaptogenesis and a reduced level of lumbar interneural activity in the extensor pool [71]. Noteworthy, Yoshida et al. recently applied three treatment modalities, namely semaphorin 3A and TMT with NSC transplantation for the chronic SCI animals, showing a significant recovery effect [72].

Griffin et al. investigated the combinatorial effect of microtubule-stabilizing drugs epothilone B and D to restrict scar-forming fibroblasts and, therefore, to reduce fibrotic scarring with bipedal and quadrupedal modalities of treadmill training. While both drugs were effectively distributed through the blood–brain barrier, epothilone B exerted a higher effect in reducing fibrotic scarring and the number of cells and axons within the lesion and increasing serotonergic fiber regeneration and VGLUT1 distal to the lesion irrespective of rehabilitation. Thus, they concluded that epothilone B and rehabilitation acted complementarily, leading to an enhanced gait recovery [73].

**Table 3 cells-13-00412-t003:** Biochemical Treatments to Counteract Regeneration Inhibitory Factors among Glial and Fibrotic Scar Components Applied with Rehabilitation.

Basic Information/Model	Rehabilitation	Combinatorial Treatment	Effect
2011 Jakeman LB [74] C57BL/6 mice (female, *N* = 16) Midthoracic spinal contusion	Subacute start at 1 week (day following initiation of medication) Voluntary wheel running exercise, for 5 weeks	Subacute; treated at 1 week -cABC (L4/5 intra-parenchymal injection, once) Groups: Wheel-run, cABC, Combined	No effect on motor and sensory function of each single treatment or combinatorial treatment.
2008 Tester NJ [56] Purpose bred SPF cats (adult, female, *N* = 9) T10 hemisection	Acute start within 2–3 days, trained before SCI (1) Bipedal TMT at 5 days/week and basic overground runway at least 3 days/week (2) A horizontal ladder, narrow beam crossing, and a pegboard 2 days/week were reintegrated when ability allowed	Immediate -cABC in gel form placed for 30 min after injury and then injected via a subdural port every other day for 1 month Groups: Training, cABC + Training	(1) While recovery of skilled locomotion (ladder, peg, and beam) was accelerated, that of basic locomotion (bipedal treadmill and overground) was unaffected; (2) Combinatorial treatment with cABC enhanced serotonergic plasticity.
2009 Garcia-Alias G [57] LH rats (male, *N* = 60), C4 dorsal funiculi cut	Subacutely started at 7 DPI, 1 h/day, 5 days/week. (1) Specific: placed in a cage where rats are facilitated to retrieve seeds with their forepaws (2) Non-specific: enriched environment cage where rats were encouraged to explore food pellets	Acute treatment -cABC or penicillinase injection above and below the lesion after SCI, followed by five intrathecal infusions on alternate days. 6 Groups: cABC or penicillinase infusion for no rehabilitation, specific or non-specific rehabilitation	(1) cABC enhanced CST axonal crossing and sprouting independently of rehabilitation regime, but specific rehabilitation showed the greatest recovery; (2) cABC + specific rehabilitation improved manual dexterity; (3) Non-specific rehabilitation improved ladder walking but showed worse skilled reaching abilities.
2011 Wang D [58] LH rats (male, *N* = 44), C4 dorsal spinal cord injury	Late-subacute start at 1 month Task-specific paw-reaching rehabilitation, 30 min twice/day for 10 weeks	Late-subacute; started at 1 month -cABC: injection rostral and caudal to the lesion + five injections (i.t.), every other day after initial treatment Groups: cABC, Rehabilitation, Combined	(1) Combinatorial treatment induced greater recovery of skilled paw reaching and ladder/beam walking; (2) Rehabilitation increased modest recovery of skilled paw reaching; (3) Combinatorial treatment increased sprouting of the CST accompanied by increased levels of VGLUT1-positive presynaptic boutons and perineuronal net component.
2016 Shinozaki M [60] SD rats (female, *N* = 61) Very severe thoracic spinal contusion	Chronic start at 6 weeks -Quadrupedal TMT, 30 min/day, 5 days/week for 8 weeks	Chronic; started at 6 weeks -cABC (i.t.) with an osmotic pump for 1 week Groups: TMT, cABC + Training	(1) Combinatorial treatment induced slight motor functional recovery; (2) Recovery at 6–9 weeks with TMT and at 12–14 weeks with cABC; (3) cABC restored the transverse residual tissue area and induced neuronal/serotonergic fiber regeneration.
2014 Alluin O [54] * Wistar rats (female, *N* = 48) Mid thoracic clip compression	Acute start on day 1 -Quadrupedal TMT, some of 15 min sessions/day, 5 days/week for 7 weeks	Acute; started on 4 DPI -cABC and growth factors (GFs: EGF, FGF2, and PDGF-AA) (i.t.) with an osmotic pump for 7 days: no single administration Groups: TMT, cABC + GFs, Combined	(1) Absence of synergistic effects on kinematic parameters after cABC + GFs treatment; (2) cABC + GFs induced synergistic effects on neuroanatomical plasticity in collateral sprouting of CST and serotonergic fibers.
2015 Ichikawa Y [64] SD rats (female, *N* = 58) C3/4 dorsal hemisection	Subacute start at 1 week -Single-pellet reaching task with Whishaw apparatus, 20 min/day, 5 days/week for 5 weeks	Subacute; started at 1 week -K-II or cABC (i.t.) with an osmotic pump for 14 days Groups: Training, K-II, cABC, K-II + Training, cABC + Training	(1) K-II combinatorial treatment induced better functional recovery than each single treatment; (2) Each combinatorial treatment synergistically increased neurite growth; (3) The effects of K-II and cABC were comparable.
2009 Maier JC [66] SD rats (female, *N* = 40 + 28), T-shaped lesion (T8), a bilateral dorsal hemisection, and a complete midline transection	Subacute start at 1 week -Bipedal TMT 20 min + quadrupedal TMT 20 min for 8 weeks	Acute intervention -Anti-Nogo-A antibody (i.t.) with an osmotic pump for 14 days Groups: TMT, anti-Nogo-A, Combined	(1) While both single treatments improved behavior, kinematics were different; (2) An anti-Nogo-A antibody increased regeneration and neuronal reorganization; (3) Combinatorial treatment did not show synergistic effects, possibly due to an increase in pain perception or interference with the recovery process.
2017 Chen K [67] SD rats (female, *N* = 28) T-shaped lesion (T9)	Subacute start at 3 weeks -Bipedal TMT 20 min + quadrupedal TMT 20 min, 5 days/week for 8 weeks	Acute intervention -Anti-Nogo-A antibody (i.t.) with an osmotic pump for 14 days -Anticyclosporin-A antibody (ACsA) as the control Groups: ACsA, TMT + ACsA, anti-Nogo-A, TMT + anti-Nogo-A	(1) Combinatorial therapy improved motor function together with CST sprouting caudal to the lesion, increased serotonergic synapses onto lumbar motoneurons, and yielded the greatest reduction of lumbar interneural activity (c-Fos expression); (2) There were no treatment effects on thermal nociception, mechanical allodynia, or lesion volume.
2013 Zhao RR [69] LH rats (*N* = 42) C4 bilateral dorsal column injuries	Late-subacute start multitask after cessation of anti-Nogo-A antibody treatment for 12 weeks (1) Montoya-type staircase reaching, 1 h/day, 5 days/week (2) Ladder walking, three rounds/session, four sessions/week	(1) Acute anti-Nogo-A (i.t.) injection for 2 weeks (2) Subacute cABC (i.t.) started at 3 weeks, five injections/10 days, with scar tissue dissection Groups: Training, Training + cABC, Training + anti-Nogo-A, Combined	(1) Each single agent in combination with rehabilitation showed similar effects, with increased sprouting and axonal regeneration; (2) Combinatorial treatment with an anti-Nogo-A antibody or cABC and rehabilitation was more effective; (3) An anti-Nogo-A antibody stimulated growth of larger axons (diameter > 3 μm), while cABC affected finer axons with varicosities more.
2014 Zhang L [71] SD rats (female, *N* = 53) T10 transection	Subacute start at 1 week -Bipedal BWS-TMT, 20 min/day, 5 days/week for 8 weeks	Immediate implantation -Silicon sheet containing semaphorin 3A inhibitor (sema3Ai) for release over 2 months Groups: sema3Ai, Combined	(1) Combinatorial treatment induced better gait kinematics than a single treatment; (2) Highest level of synaptogenesis and reduction of lumbar interneural activity in the extensor pool at L4.
2023 Griffin JM [73] SD rats (*N* = 194) T10 Moderate contusion.	Subacute start at 3 weeks post SCI -Bipedal and quadrupedal training. 20 + 20 min/days for each training, 5 days/week for 7 weeks	Acute -Epo D, Epo B, or Ixabepilone at 1 and 15 DPI (s.c. or i.p.) Groups: Each medication, Training, Epo D, Epo D + Training, Epo B, Epo B + Training	(1) Ixabepilone did not effectively cross the blood–brain barrier; (2) Epo B exerted the highest effect to decrease fibrotic scarring and increase serotonergic fiber regeneration and VGLUT1 expression; (3) Epo B and rehabilitation acted complementarily on gait parameter.

BWS—body-weight-supported; cABC—chondroitinase ABC; CST—corticospinal tract; DPI—days postinjury; EGF—epidermal growth factor; Epo—epothilone; FGF2—fibroblast growth factor 2; LH—Lister hooded; K-II—keratanase II; PDGF-AA—platelet-derived growth factor-AA; SCI—spinal cord injury; SD—Sprague–Dawley; TMT—treadmill training. *: a study also included in another table.

### 3.4. Other Biochemical Treatments

In the Table 4, the other pharmacological agents used in combination with training are summarized in alphabetical order. Although functional recovery was reported, the mechanisms were not always investigated. Many proved neuroprotective, but some seemed to induce neuroregenerative effects. The variety of studies in this section include a study that targeted the musculoskeletal system with testosterone [75], a study that targeted urination [76], and a series of studies utilizing induced low-grade inflammation [77,78]. In this section, we review them according to the representative mechanisms.

#### 3.4.1. Neuroprotective Agents

The following studies investigated specific treatment mechanisms. Goldshmidt et al. used blood glutamate scavengers and showed that single treatment decreased the level of glutamate in cerebrospinal fluid and increased axonal survival and GAP-43 expression in neuronal cells. In addition, they showed that combinatorial treatment reduced inflammation, scarring, and lesion size; enhanced axonal regeneration throughout the lesion; increased the level of synapses around motor neurons; and subsequently improved motor function [79]. Liu et al. investigated the effect of nutritional therapy with omega-3 polyunsaturated fatty acid and docosahexaenoic acid, which was reported to promote neuroplasticity after SCI. They showed that combinatorial therapy induced better functional recovery in accordance with more sprouting of uninjured CST and serotonergic fibers as well as synaptogenesis [80]. Melatonin has attracted attention for its beneficial effects on nitric oxide synthesis and on adult stem cells independently of circadian regulation, also inducing neural regeneration. Park et al. showed that melatonin restored hindlimb movement and the number of motor neurons in the ventral horn and reduced the level of inducible nitric oxide synthase (iNOS) [81]. Lee et al. showed that melatonin administration from the early phase after SCI increased dendritic spine density, and combinatorial therapy facilitated hindlimb functional recovery in line with reduced lesion size, increased density of dendritic spines and axons, and increased number of BrdU- and nestin-positive endogenous NSCs [82]. Osuna-Carrasco et al. investigated the combinatorial effects of tamoxifen, a selective estrogen receptor modulator. While tamoxifen and training preserved the lesion tissue or induced better morphology, the combinatorial treatment yielded the best kinematic results [83]. Liu et al. used methylprednisolone as a neuroprotective agent in acute SCI rats and applied it in combination with TMT in the subacute phase. They found that the combinatorial treatment induced better hindlimb function. They reported more preserved tissue and reduced expression of the Nogo receptor and CSPGs at the lesion [84]. To restore the pre- and post-synaptic inhibition to mitigate spasticity, Caron et al. investigated the effect of bumetanide, a sodium-potassium-chloride intrude (NKCC1) antagonist, to restore post-synaptic inhibition. While they did not combine bumetanide with rehabilitation as a long-term treatment, they found presynaptic inhibition was decreased only when bumetanide was acutely administered to step-trained animals; thus, bumetanide and training are potentially competing [85].

#### 3.4.2. Neuroregenerative Agents

Wei et al. investigated the effect of inhibiting cortical PKA using intrathecal Rp-cAMPS to increase the level of cAMP, which is associated with neuron sprouting and neurite extension. They showed that combinatorial treatment with Rp-cAMPS and rehabilitation promoted functional recovery and collateral sprouting of CST axons. In addition, they showed that Rp-cAMPS does not affect CREB phosphorylation [86]. Yin et al. applied a Taxol-modified collagen scaffold, with both Taxol and the collagen scaffold possessing neuroprotective properties, to thoracic-cord-transected dogs. While the treatment effect of each component was not evaluated, they found that the implantation promoted motor recovery in accordance with electrophysiological and histological recovery, including neurogenesis, axonal regeneration, and reduced glial scar formation [87].

#### 3.4.3. Others

In terms of other functional targets, Yarrow et al. performed adjuvant weekly testosterone injections in combination with TMT to modify bone and muscle metabolism. While single treatment with testosterone suppressed bone resorption, attenuated cancellous bone loss, and limited muscle fiber-type transition and atrophy, along with a 20% recovery in gait, the addition of training stimulated bone formation and maintained muscle force production along with a 75% recovery in gait [75]. Hubscheter et al. used desmopressin, a synthetic analog of arginine vasopressin that is used to treat nocturnal polyuria in SCI patients, in combination with TMT. They reported that each single treatment effectively decreased urination despite there being no additive or synergistic effects [76].

Wong et al. and Krisa et al. independently reported almost opposite effects when amphetamine, which enhances the effects of rehabilitation in stroke and brain injury models [88], was applied to subacute SCI animals. Krisa et al. found that combinatorial treatment qualitatively improved locomotion, while there was no evidence of neuroprotection. It is noteworthy that rehabilitation was performed more frequently at high intensity as task-specific forelimb motor training for long periods in the study by Krisa et al. [88,89]. Because training has a threshold to elicit combinatorial effects [8], these results imply that the training load was lower than this threshold in the study by Wong et al.

While the present review does not include the scaffold single treatment, it is noteworthy that researchers utilize the bioreactivity of some of the biomaterials as a scaffold, as represented by plasma-synthesized polypyrrole/iodine (PPy/I). It was reported that PPy/I reduces inflammatory response and promotes neural regeneration. While treadmill training did not show a remarkable combinatorial effect, a Mexican group combined PPy/I nanoparticles with a mixed rehabilitation scheme comprising swimming training and an enriched environment. They reported that neural tissue preservation and gene expression related to proliferation, cell development, morphogenesis, cell differentiation, neurogenesis, neuron development, and synapse formation process were explicitly superior in the combinatorial treatment [90,91,92].

Contrary to those pharmacological interventions, Fouad’s group used single-pellet reaching training in combination with systemic injection of low-dose lipopolysaccharide to induce inflammation in chronic and subacute SCI. They found that compensatory strategies (restoration of original function rather than acquisition of new motor strategies) were not involved in motor recovery. Combinatorial treatment promoted recovery of the cortical drive to affected forelimb muscles and the restructuring of corticospinal innervation in the chronic phase [78]. On the other hand, they found that lipopolysaccharide paradoxically suppressed neuroinflammation at the lesion and increased long-term anxiety-like behavior in the subacute model [77].

**Table 4 cells-13-00412-t004:** Combinatorial Treatment with Other Agents.

Basic Information/Model	Rehabilitation	Combinatorial Treatment	Effect
2012 Wong JK [89] SD rats T9 hemisection (female) C5 hemisection (male)	Acute (main) or subacute start at 1 or 14 DPI -Two trials of a beam walking task for 5 days over 10 days (every other day)	Paired with training -AMPH (i.p.) Groups: Training, Combined	(1) AMPH treatment in combination with testing/retraining resulted in a larger lesion and impaired locomotor recovery; (2) Results were unique in the cervical and thoracic SCI models regardless of the treatment phase.
2012 Krisa L [88] SD rats (male, *N* = 101) C3–C4 right side contusion	Subacute start at 13 DPI -Task-specific forelimb motor training, 15 min twice/day, 7 days/week for 12 weeks	Together with training -AMPH (i.p.) every third day until week 12 of training, excluding testing weeks Groups: Training, AMPH, Combined, Combined + EE	(1) Combinatorial treatment improved qualitative reaching but not kinematics, and there was no evidence of neuroprotection; (2) Rats in EE combined group showed less improvement.
2020 Goldschmit Y [79] C57BL/6 mice (male, *N* = 82) T12 left hemisection	Subacute start at 1 week -Quadrupedal TMT with textured tread, 10 min twice/day, 5 days/week for 3 months	Acute; 60 min after SCI -BGS, rGOT1, and OxAc (i.v.) for 5 consecutive days Groups: TMT, BGS, Combined	(1) BGS treatment decreased the level of glutamate in CSF and increased axonal survival and GAP-43 expression in neuronal cells; (2) Combinatorial treatment reduced inflammation, scarring, and lesion size; enhanced axonal regeneration through the lesion; increased the level of synapses around motor neurons; and improved motor function at 3 months post SCI.
2022 Hubscher CH [76] Wistar rats (male, *N* = 60) T9 moderate contusion	Subacute start at 2 weeks -Activity-based forelimb training + TMT stepping, 70 daily sessions	Chronic medication after all training sessions -Desmopressin (DDAVP, synthetic analog of arginine vasopressin) (i.p.) 3 days Groups: Training, DDAVP, Combined	Either intervention or treatment alone effectively decreased urination despite there being no additive effect.
2017 Liu ZH [80] SD rats (female) C4/5 lateral hemisection	Acute start at 2 DPI -Pellet grasping training, 30 min twice/day for 3 weeks	Immediate; 30 min after SCI -DHA (i.v.) Groups: Training, DHA, Combined	(1) Combinatorial therapy induced greater functional recovery; (2) Combinatorial therapy induced more sprouting of uninjured CST and serotonergic fibers and synaptogenesis.
2018 Torres-Espin A [78] Lewis rats (female, *N* = 132) C4 dorsolateral quadrant section	Chronic start at 8 week -Forelimb single-pellet reaching and grasping, 10 min/session, 4–5 days/week	Chronic inflammation induction at 8 weeks -Low dose of LPS (i.p.) Groups: Training, LPS, Combined	(1) Rehabilitation enhanced recovery dependent on intensity; (2) Combinatorial treatment restored original function rather than enhancing new motor strategies; (3) Cortical drive to affected forelimb muscles was recovered, and corticospinal innervation was restructured.
2021 Schmidt E [77] Lewis rats (female, *N* = 60) C4 dorsolateral quadrant section	Subacute start at 14 DPI -Forelimb single-pellet reaching and grasping at high intensity, 10 min/session, 4–5 days/week	Subacute inflammation induction at 10 DPI -LPS (i.p.) Groups: Training, Combined	(1) Motor function was recovered secondarily without compensatory strategies; (2) Combinatorial treatment with LPS resolved chronic neuroinflammation around the lesion; (3) Combinatorial treatment with LPS increased long-term anxiety-like behavior.
2021 Liu JT [84] SD rats (male, *N* = 50) T10 moderate contusion	Subacute start at 2 weeks -Quadrupedal TMT, 10 min twice/day, 6 days/week for 8 weeks	Acute; 30 min after SCI -MP (i.v.) Groups: MP, Combined	Combinatorial treatment induced (1) Better H/L function with preserved histology; (2) Reduced expression of Nogo receptor and CSPGs.
2010 Park K [81] SD rats (male, *N* = 24) -T10 moderate contusion	Acutely started at 3 DPI -TMT for 15 min, twice a day, 6 days/week for 4 weeks	Acutely started at 1 DPI -MT administration twice per day for 28 days Groups: TMT, Combined	Combinatorial treatment with MT increased hindlimb movement and number of motor neuron at ventral horn and reduced level of iNOS.
2014 Lee Y [82] SD rats (male, *N* = 21) T10 moderate contusion	Acute start at 3 DPI -TMT for 15 min, 6 days/week for 18 days	Immediate -MT administration twice per day for 21 days Groups: MT, Combined	(1) MT increased dendritic spine density; (2) Combinatorial treatment facilitated H/L functional recovery, reduced lesion size, increased the density of dendritic spines and axons, and increased the number of BrdU- and nestin-positive endogenous NS/PCs.
2016 Wei D [86] Lewis rats (female, *N* = 23 + 20) C4 unilateral dorsolateral quadrant lesion	Subacute start at 1 week -Single-pellet skilled reaching task, 15 min/day, 5 days/week for 6 weeks	Immediate -Inhibition of cortical phosphokinase A using Rp-cAMP (i.t.) with an osmotic pump for 6 (experiment 1) or 4 (experiment 2) weeks Groups: Training, Rp-cAMP, Combined	Combinatorial treatment with Rp-cAMPS promoted functional recovery and collateral sprouting of CST axons, and Rp-cAMPS did not affect CREB phosphorylation.
2016 Osuna-Carrasco LP [83] SD rats (female, *N* = 24) Penetrating ventral injury T13–L1	Acute start at 3 DPI -TMT, 5 days/week, increasing duration (10–20 min for 4 weeks)	Immediate: 0, 24, and 48 h after SCI -TMX (a selective estrogen receptor modulator) (i.p.) Groups: TMT, TMX, Combined	(1) Combinatorial treatment induced the best kinematics; (2) TMX preserved spinal cord gray and white matter; (3) Training improved morphology.
2018 Yin W [87] Beagle canines (18 months old, female, *N* = 12) T8 transection	Acute start -Passive ROM exercise Subacute -Ambulation training with slings and wheelchairs to support hip joints	Immediate implantation -Neuroprotective Taxol-modified LOCS Groups: Training, LOCS + Taxol	LOCS + Taxol promoted locomotor recovery with MEP improvement, neurogenesis, and axon regeneration to reconnect the spinal cord stumps and reduced glial scar formation.
2020 Yarrow JF [75] SD rats (*N* = 73) T9 severe contusion	Subacute start at 7 DPI -Quadrupedal manually assisted BWS-TMT, 40 min/day, 5 days/week	Acute -Adjuvant TE weekly (i.m.) Groups: TE, Combined	(1) TE suppressed bone resorption, attenuated cancellous bone loss, and constrained muscle fiber-type transition and atrophy, and stepping was recovered in 20% of rats; (2) Addition of training stimulated bone formation and maintained muscle force production, and stepping was recovered in 75% of rats.

AMPH—amphetamine; BDNF—brain-derived neurotrophic factor; BGS—blood glutamate scavenger; BWS—body-weight-supported; CSF—cerebrospinal fluid; CSPG—chondroitin sulfate proteoglycan; CST—corticospinal tract; DHA—docosahexaenoic acid; DPI—days postinjury; EE—enriched environment; H/L—hindlimb; iNOS—inducible nitric oxide synthase; IPSP—inhibitory postsynaptic potentials, KLF6—Krüppel-like factor 6; LOCS—linear-ordered collagen scaffold; LPS—lipopolysaccharide; MEP—motor-evoked potential; MP—methylprednisolone; MT—melatonin; NT3—neurotrophin 3; NS/PC—neural stem/progenitor cells; PPy/I—polypyrrole/iodine; SCI—spinal cord injury; SD—Sprague–Dawley; TE—testosterone; TMT—treadmill training; TMX—tamoxifen; VGLUT1—vesicular glutamate transporter 1.

## 4. Discussion

The present review summarizes the agents used in combination with rehabilitation and highlights the immaturity of this research area and current issues. While not many studies have investigated the mechanisms of combinatorial effect, we can characterize them into neuroprotection, neuroplasticity, and neural regeneration as assessed by histological means and neuroconductivity by electrophysiology. However, studies on each single drug were too few to clarify the total mechanisms.

### 4.1. Mechanisms Underlying the Beneficial Effects of Combinatorial Treatment

Regarding neuromodulatory treatments, most studies used 5-HT agonists. Some studies reported neural regeneration and plasticity in the serotonergic system when these treatments were performed in combination with training [30,34,38]. Others reported beneficial effects in terms of metabolism [35], electrophysiology [37], and muscle morphology [39]. Notably, some researchers suggest that serotonergic treatment is not always collaborative [27] or sometimes exacerbates spasticity [38]. To optimize the combinatorial effect, van den Brand et al. proposed a new rehabilitation paradigm to apply different trainings according to the stage of recovery. Their approach is to first improve the functionality of lumbosacral circuits with TMT supported by a combination of medication and electrical stimulation for animals with severe SCI. When the voluntary gait is observed, the training regimen is gradually switched to a complex one for better functional recovery: firstly bipedal overground gait training with robotic support and, finally, obstacles and stairs, which require voluntary gait modification [30].

BDNF and NT3 are the most commonly used neurotrophic factors in rehabilitation training. The beneficial effects of combinatorial treatment are characterized as neuroprotection and neural regeneration including myelination in the studies using BDNF when applied with a biofunctional scaffold [44]. On the other hand, NT3 induced neural regeneration, neuroprotection, and spinal neuron and perineuronal net remodeling. In addition, NT3 further restored KCC2 expression by suppressing the hyper-excitatory state of the injured spinal cord [47,93] and induced an increased electrophysiological response and retrograde tracer transport [51]. When combined with rehabilitative training, GDNF and G-CSF induced neuroprotection [52,53], and a cocktail of neurotrophic factors induced neural regeneration when applied with cABC [54]. Because rehabilitative training, especially physical exercise, upregulates these neurotrophic factors regardless of chronicity [94], the treatment effect of trophic factor(s) will be enhanced when combined with training. In addition, Liu et al. revealed that IGF-1 together with osteoponin treatment via AAV restores CST axon-dependent forelimb functions [95], and Hwang et al. clarified that the IGF-1 receptor plays a critical role in the beneficial effect of TMT on transplanted NSCs [96]. These studies suggest that a variety of neurotrophic factors and rehabilitative trainings have synergistic effects in neuroprotection, neural regeneration, as well as restoration of neuroconductivity.

Among pharmacological agents to degrade glial scars or to inhibit axonal regeneration inhibitor, it seems that neural system remodeling and regeneration are the chief mechanisms of the combinatorial effect with rehabilitative trainings. It was shown that cABC enhanced skilled motor function [56,58], in line with neural regeneration, as demonstrated by sprouting of CST and VGLUT1-positive presynaptic boutons, upregulation of the perineuronal net component [58], and serotonergic plasticity [56]. Keratanase II showed synergistic effects with training on neurite growth [64]. A semaphorin 3A inhibitor also elicited a synergistic effect on gait appearance, increased synaptogenesis, and reduced lumbar interneural activity in the extensor pool at the lower-lumbar enlargement [71]. On the other hand, anti-Nogo-A antibody treatment does not always induce a synergistic effect with rehabilitative interventions [66]. A recent four-arm study showed neural system remodeling effects, including CST sprouting caudal to the lesion, increased serotonergic synapse formation on lumbar motoneurons, and the greatest reduction of lumbar interneural activity. In addition, an anti-Nogo-A antibody and cABC were reported to induce more significant effects in terms of sprouting [69].

On the other hand, the present review found that research of immunomodulatory agents is particularly insufficient. Methylprednisolone and blood glutamate scavengers were the only agents whose combinatorial effects were studied [79,84,97]. These studies revealed that the immunomodulatory agents protect tissues, inhibit inflammation, and suppress scar formation and thus may be good candidates for further investigation. Although not a rehabilitative intervention, Huie et al. investigated the effect of TNF-α and its blockade in a spinal learning task model. They found that TNF-α inhibited spinal learning, while a TNF-α inhibitor protected against inhibition of spinal learning via intermittent peripheral stimulation [98].

In Figure 1, we summarize the additive, complementary, or synergistic effect of training across the types of medication. Many researchers reported negative results regarding gross tissue histology. A few studies reported tissue sparing, including suppression of cavity expansion or lesion extension, in combination with G-CSF and tamoxifen [52,83], and one study reported a positive trend with a growth factor cocktail [54], but studies on amphetamine, blood glucose scavenger, PKA inhibitor, and melatonin reported negative results [79,82,86,89]. Reporting bias might be involved because this assessment is fundamental. Facilitation of CST regeneration, as assessed by BDA tracing, was broadly reported in neuromodulatory treatment with quipazine, 8-OH-DPAT, D1 agonist, and electrical stimulation [14,30]; anti-Nogo-A treatment [66,67]; combinatorial treatment with cABC and GFs [54]; neuroprotective treatment with DHA [80]; and low-grade inflammation treatment with LPS [78]. While CST regeneration has two aspects, namely regeneration to the lower cord through the scar and collateral sprouting above the lesion, it seems dependent on the medical agent [54,67]. Facilitation of 5-HT fiber regeneration is one of the other chief mechanisms. Some researchers reported positive effects using different modalities: TMRD (tetramethyl rhodamine dextran) axonal tracing of descending axon [79] and NF200 in the scaffold [47].

### 4.2. Rehabilitative Training and Pharmacological Treatment Sometimes Act as Competing

Although many pharmacological agents act additively, complementarily, and synergistically with rehabilitation, no collaborative effects have been reported in some cases, and the effects seem to compete in some cases. Among the studies that only investigated behavioral assessments, quipazine and robot-assisted bipedal TMT [28] and fluoxetine and TMT [37] lacked collaborative effects. In a study that combined bipedal and quadrupedal TMT with anti-Nogo-A treatment, no synergistic effects were observed [66]. The combinations of dopaminergic treatment and passive motorized bicycles, fluoxetine and TMT, and cyproheptadine and TMT did not show any collaborative effect on spasticity, as assessed by behavior and electrophysiological evaluation [38,40]. In addition, although every single treatment with DD-AVP medication and TMT showed a beneficial effect on urination, their combinatorial treatment did not have any additive effect [76].

Maier et al. claimed the reason why combinatorial treatment lacked collaborative effect is possibly due to an increase in pain perception [66]. Caron et al. showed a competing effect of training with NKCC1 antagonists in the suppression of presynaptic inhibition [85]. If presynaptic inhibition decreases, pain-related behavior may increase due to the loss of suppression in the spinal circuit. This might have a mechanism similar to that of passive bipedal TMT-exacerbated hyperalgesia [99], whereas voluntary bipedal TMT suppressed hyperalgesia [49]. Thus, it was unclear whether the medication or rehabilitation contributed to the exacerbation of pain perception in the former case (Table 5).

Furthermore, some studies reported worse outcomes with combinatorial treatment using non-specific rehabilitation regimens. First, when Foffani compared the combinatorial effect of quipazine with bike training and TMT, the responding cell number in the somatosensory cortex was increased in quipazine with TMT while decreased in that with bike training in neurophysiological assessment. Thus, the outcome was favorable in the quipazine and TMT group [27]. Second, while cABC combined with specific forelimb rehabilitation showed the greatest recovery, cABC combined with non-specific rehabilitation resulted in worse skill-reaching abilities. However, this case appears reasonable because the beneficial effect is related to how close the rehabilitation content is to what was assessed. Indeed, the nonspecific rehabilitation group showed better motor functional ability in the ladder walking test [57]. Third, whereas Krisa et al. found amphetamine and task-specific forelimb training to induce synergistic recovery, the effect reversed when the animals in a combination group were kept in an enriched environment [88]. In addition, another study also reported the competing effect of acute beam walking and amphetamine treatment [89]. These studies suggest that medical treatment can promote maladaptive plasticity due to a less-specific and wrong rehabilitation strategy. Since researchers who are not clinicians tend to misunderstand that all exercise training programs have similar characteristics, these studies will be very important in showing the competition between rehabilitative training and medication in some cases.

### 4.3. Prospects of Rehabilitation Training and Medication for SCI Treatment

The development of treatments for chronic SCI, which exerts refractory characteristics onto treatment, is very important because there are 47 times more patients with chronic SCI than with acute-to-subacute SCI [60]. Treatment resistance of the chronically injured spinal cord is characterized by persistent low-grade inflammation, fibrotic tissue remodeling, and glial scar formation around the lesion site [20], all of which can be targeted by pharmacotherapy. Rehabilitation training is usually feasible and noninvasive and does not pose ethical problems in the clinical setting. In addition, it induces neural plasticity through functional training, upregulates expression of neurotrophic factors through physical exercise, and resolves deconditioning in chronic SCI [8]. These properties of rehabilitation will enhance the effect of pharmacotherapy. However, the current review found that only a few studies targeted the chronically injured spinal cord, including those using cABC [60], desmopressin [76], and lipopolysaccharide [78]. Only two further studies were added when studies for the late-subacute phase were included: fluoxetine or cyproheptadine [38] and cABC [58]. This underlies the importance of further investigation into pharmacotherapy in combination with for the chronically injured spinal cord.

Although drug–drug combination is another treatment strategy, only a few investigators have attempted this approach together with rehabilitation training. While limited effects of BDNF and NT3 pairings were reported in a AAV study [100], researchers showed that treatment with a combination of buspirone, carbidopa, and L-DOPA [39] and cABC and an anti-Nogo-A antibody [69] or growth factors [54] induced significantly greater recovery. It is noteworthy that there are no investigations about medicines classified into the “others” domain. Drug pairs that have completely different sites of action tend to elicit synergistic or additive effects. While some agents compete with a particular type of training in specific conditions [27], further combinatorial treatments whose effects are different to those of rehabilitation are good candidates for clinical treatment of SCI.

While the present review does not included viral or gene treatments, the following treatment effects have been studied: chemogenic activation of propriospinal neuron with non-invasive intravenous AAV9 delivery [101], KLF6 expression to enhance CST axon sprouting [102], PTEN inhibition to enhance CST axon sprouting and synapse reformation [103], PTEN/SOCS3 co-inhibition to promote axonal regeneration [104], ADAMTS4 induction to decrease lesion size [63], MMP-9 deletion to attenuate remote microglial activation and restore TNF-α expression [97], and blocking of NB-3 (contactin-6) to promote synapse reformation between serotonergic Raphe Spinal tract regenerative axons and motor neurons [105]. These factors could be potential targets of medical treatment. Physical medicine modalities including EES, EMS, transcranial current stimulation, and transcranial magnetic stimulation [106] and cell therapies such as transplantation of NSCs, mesenchymal stem cells, and OECs [8] are the other treatment modalities to study in combination. While the importance of such an inter-disciplinary approach is emphasized [107], few researchers have investigated the effect of these treatments on chronic SCI in preclinical studies, although there are a number of human studies. Leydeker et al. used the combination of trans-spinal magnetic stimulation and aerobic exercise [108] in the research field of physical medicine, Theisen et al. used the combination of chronic peripheral nerve grafting and rehabilitation training [109], and Tashiro et al. and Shibata et al. investigated the combinatorial effect of chronic NSCs transplantation with TMT in terms of locomotor and sensory function [25,110,111] in the field of regenerative medicine. However, all these attempts failed to achieve functional recovery significantly superior to single rehabilitation treatment. A recent study reported that further combination of a semaphorin 3A inhibitor significantly enhanced the treatment effect of combined treadmill training and NSCs transplantation [72]. Based on these findings, a combinatorial strategy incorporating more than two factors may be needed.

### 4.4. Limitation

Since the current review is qualitative, we cannot distinguish which combination is more beneficial than multiple or single treatments. It was difficult to compare the effect sizes due to heterogeneity in species, models, chronicity, rehabilitative interventions, and assessments as well as the lack of quality and the low quantity of the studies. Importantly, many of the studies did not elucidate or even investigate the mechanisms underlying the phenomena. In addition, a large portion of the studies included in the present review used rodents, whose spinal neuronal system is different from that of humans, and therefore, the findings should be interpreted with caution [112]. In addition, we must consider the shortage of rehabilitation animal models. Rehabilitation training is not very feasible in preclinical studies and is time-consuming and expensive, and the appropriate training and its load have not been structurally validated or standardized among research groups [20]. Standardized protocols have only very recently been proposed for forelimb functional training and quadrupedal TMT in SCI model rodents [94,113]. The present review facilitates a fair comparison of the agents because old studies often had a preliminary design; e.g., they lacked a control or combination group, displayed heterogeneity of intervention conditions, and had an insufficient sample size.

While rehabilitative training and physical medicine treatments are usually applied together in SCI patients, the single treatment effect of physical medicine is often studied in the preclinical studies. In this review, our emphasis is specifically directed towards rehabilitative training, excluding studies solely investigating physical medicine single treatments. This decision is rooted in the recognition of the unique attributes of rehabilitative training, which actively engages the impaired neuronal system through task-oriented activities. However, we acknowledge the significance of exploring the combined effects of physical medicine and pharmacological treatments, which hold relevance for both researchers and clinicians in the SCI field. Therefore, we intend to consider reviewing the combined effects of physical medicine and pharmacological treatments to provide comprehensive insights into the treatment landscape for SCI.

## 5. Conclusions

The present review revealed that the combinatorial effects of a variety of medical agents with rehabilitative training have been investigated. However, a large portion of the mechanisms remains to be elucidated because many of the studies only described functional aspects. In addition, studies targeting the chronically injured spinal cord are rare, although a combinatorial strategy is much more critical in this scenario. Further investigations are needed to elucidate the mechanisms underlying the beneficial effects and to develop a strategy for treating the refractory chronically injured spinal cord.

## Figures and Tables

**Figure 1 cells-13-00412-f001:**
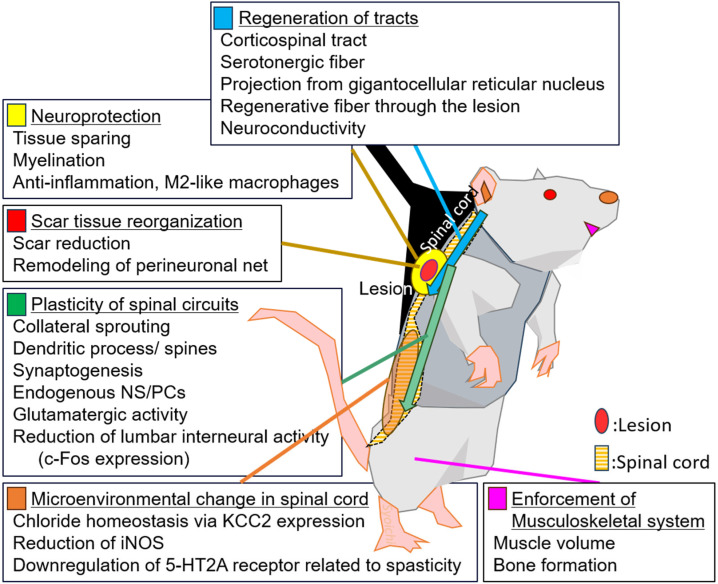
Summary of combinatorial effects of rehabilitative training. The combinatorial effects of rehabilitative training are illustrated. Beneficial effects are classified as regeneration of tracts, neuroprotection, scar tissue reorganization, plasticity of spinal circuits, microenvironmental change in the spinal cord, and enforcement of the musculoskeletal system. iNOS—inducible nitric oxide synthase; NS/PC—neural stem/progenitor cells; KCC2—kalium-chloride cotransporter 2.

**Table 5 cells-13-00412-t005:** Competing Effects with Pharmacological Treatment and Rehabilitative Training.

Agent	Training	Treatment Effect	Condition, Study
Quipazine	TMT	Increase in responding cells’ number in somatosensory cortex	Subacute, SD rats, T8/9 transection [27]
	Bike training	Decrease in responding cells’ number in somatosensory cortex	
cABC	Specific training with seeds retrieving	Better skill reaching accompanying histology	Subacute, LH rats, C4 dorsal funiculi cut [57]
	Enriched environment	Worse skill reaching	
Amphetamine	Single-pellet test training Staircase test training	Better skill reaching	Subacute, SD rats, C3–4 hemicontusion [88]
	To further add enriched environment	Worse skill reaching	
	Beam walking	Impaired locomotion	Acute, SD rats, 2 models [89]

cABC—chondroitinase; LH—Lister hooded; SD—Sprague–Dawley; TMT—treadmill training.

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
