# Peer review of "Do Pharmacological Treatments Act in Collaboration with Rehabilitation in Spinal Cord Injury Treatment? A Review of Preclinical Studies"

_cells, 2024, doi:10.3390/cells13050412_

Round 1

Reviewer 1 Report

Comments and Suggestions for Authors

In this review, the authors present the studies that have investigated the effect of Pharmacological Treatments combined with Rehabilitation in comparison to treatment or rehabilitation itself. This is an interesting perspective and the review is well-structured and well written as a whole. An important work has been done to provide a good overview of the work done until now.

We can however regret that the introduction is too broad (second part). Please provide instead a structured statement showing why this review article is important considering the field, its significance, and main mechanisms on which it aims to focus. Also, one would highlight what makes this review article is unique?

Also, I suggest the authors to add columns in their Table 5 corresponding to the type of pharmacological treatments that change the effect of training. they would report positive (or negative) impact of the combination of treatment by adding a +, = or – when studies report such effect in front of the mechanisms proposed. It would add a lot to the review, providing a sort of summary of the findings.

Lastly, one limitation should be added regarding quantitative assessments. Indeed, this is a qualitative review that does not provide any information on the extent to which it is more beneficial to combine treatments.

Minor comments:

Abstract lie 26, “further combinations of these agents” unclear, please clarify

Introduction: line 77-79 “broad sense” is repeated twice

As a whole, the introduction should be improved as suggested above

Results: line 155: any explanation?

Line 216: unclear, please clarify

Comments on the Quality of English Language

Overall good quality. Please go through the manuscript to do sime minor editing of English language. Some repetitions and some missing words.

Author Response

Reviewer 1

In this review, the authors present the studies that have investigated the effect of Pharmacological Treatments combined with Rehabilitation in comparison to treatment or rehabilitation itself. This is an interesting perspective and the review is well-structured and well written as a whole. An important work has been done to provide a good overview of the work done until now.

We can however regret that the introduction is too broad (second part). Please provide instead a structured statement showing why this review article is important considering the field, its significance, and main mechanisms on which it aims to focus. Also, one would highlight what makes this review article is unique?

(Response) We truly thank the reviewer providing valuable comments to improve our draft. Concerning “the introduction, second part”, we reorganized the corresponding part to emphasize why this review is important together with main mechanisms. In addition, we moved the latter half of the part describing why we did not include physical medicine modalities to the limitation section. Concerning the regenerative medicine modalities, we decided to delete it because other combinatorial treatment modalities like viral therapy, scaffold etc., did not have such explanation.

Combinatorial treatment to enhance the effects of rehabilitation is another prominent theme of preclinical studies in SCI research. While rehabilitation induces various beneficial changes, as represented by the promotion of neuronal plasticity and regeneration and modification of microenvironments in the injured spinal cord on its own, and these changes are directly linked to functional recovery, additional treatments may support and enhance the effect. Pharmacological treatments, including medication, biological agents such as neurotrophic factors and biomaterials, cell transplantation, and physical medicine modalities, will be good candidates because it is the most common therapeutic strategies for diseases and has advantages in terms of their non-invasiveness, feasibility, and ethicality. However, to our knowledge, there are no reviews regarding combinatorial therapy with a variety of medical agents and rehabilitation. Although this research field has a relatively long history, relatively few studies have fairly evaluated the effects of combinatorial treatment. Some early studies seem to have an insufficient design, such as a lack of appropriate controls, heterogeneity of intervention conditions, a lack of combinations, and an insufficient sample size. This fact has made it difficult for the researchers to distinguish which combinatorial effect might be significant in truth. In addition, the difference of treatment effect, that are rooted on the strategy of rehabilitative training, is sometimes overlooked by non-clinician researchers, even though some combinations have shown competing effects. On these grounds, we first summarized combinatorial treatments with rehabilitation and medication. In addition, we focus also on the cases in which these two modalities showed competition. This review will help facilitate preclinical and clinical research on rehabilitation and medication and further develop SCI treatments. (lines 69-90)

While rehabilitative training and physical medicine treatments were usually applied together to the SCI patients, the single treatment effect of physical medicine is often studied in the preclinical studies. In this review, our emphasis is specifically directed towards rehabilitative training, excluding studies solely investigating physical medicine single treatments. This decision is rooted in the recognition of the unique attributes of rehabilitative training, which actively engages the impaired neuronal system through task-oriented activities. However, we acknowledge the significance of exploring the combined effects of physical medicine and pharmacological treatments, which hold relevance for both re-searchers and clinicians in the SCI field. Therefore, we intend to consider reviewing the combined effects of physical medicine and pharmacological treatments to provide com-prehensive insights into the treatment landscape for SCI. (Lines 666-676)

Also, I suggest the authors to add columns in their Table 5 corresponding to the type of pharmacological treatments that change the effect of training. they would report positive (or negative) impact of the combination of treatment by adding a +, = or – when studies report such effect in front of the mechanisms proposed. It would add a lot to the review, providing a sort of summary of the findings.

(Response) We truly appreciate this reviewer for this advice to improve the quality of discussion of current review. While the Table 5 listed more generalized effect of trainings, we noticed that the competing effects were reported in more specific cases. Those were so specific that the non-favorable effect is dependent on rehabilitation strategy and observed only in specific outcome measurements. Therefore, we decided to make a new Table to explain it. In this process, we changed the order of competing pairs according to the order of appearance in the Result section, and added a few explanatory sentences to promote the reader’s understanding.

Table 5. Competing Effects with Pharmacological Treatment and Rehabilitative Training.

Agent

Training

Treatment effect

Condition, Study

Quipazine

TMT

Increase in responding cells number in somatosensory cortex

Subacute, SD rats, T8/9 transection [30]

Bike training

Decrease in responding cells number in somatosensory cortex

cABC

Specific training with seeds retrieving

Better skill reaching accompanying histology

Subacute, LH rats,
C4 dorsal funiculi cut [56]

Enriched environment

Worse skill reaching

Amphetamine

Single-pellet test training

Staircase test training

Better skill reaching

Subacute, SD rats, C3-4 hemicontusion [87]

To furthre add Enriched environment

Worse skill reaching

Beam walking

Impaird locomotion

Acute, SD rats, 2 models [88]

cABC: chondroitinase; LH: Lister hooded; SD: Sprague-Dawley; TMT: Treadmill training.”

Furthermore, some studies reported worse outcomes with combinatorial treatment using non-specific rehabilitation regimens. First, when Foffani compared the combinatorial effect of quipazine with bike training and TMT, the responding cell number in the somatosensory cortex was increased in quipazine with TMT while decreased in that with bike training in neurophysiological assessment. Thus, the outcome was favorable in the quipazine and TMT group [27]. Second, while cABC combined with specific forelimb rehabilitation showed the greatest recovery, cABC combined with non-specific rehabilitation resulted in worse skill-reaching abilities. However, this case appears reasonable because the beneficial effect is related to how close the rehabilitation content is to what was assessed. Indeed, the nonspecific rehabilitation group showed better motor functional ability in the ladder walking test [57]. Third, whereas Krisa found amphetamine and task-specific forelimb training induce synergistic recovery, the effect reversed when the animals in a combination group were kept in an enriched environment [88]. In addition, another study also reported competing effect of acute beam walking and amphetamine treatment [89]. These studies suggest that the medical treatment can promote maladaptive plasticity due to less-specific wrong rehabilitation strategy. Since researchers who are not clinicians tend to misunderstand that all exercise training programs have similar characteristics, these studies will be very important in showing the competition between rehabilitative training and medication in some cases. (lines 579-597)

 Lastly, one limitation should be added regarding quantitative assessments. Indeed, this is a qualitative review that does not provide any information on the extent to which it is more beneficial to combine treatments.

(Response) We agree with the reviewer on this comment as well. While this point was mentioned in the beginning of the section, we clearly stated the limitation as the reviewer indicated at the top of limitation section.

“Since the current review is qualitative, we cannot distinguish which combination is more beneficial than others or the single treatments.” (lines 650-651)

Minor comments:

Abstract line 26, “further combinations of these agents” unclear, please clarify

(Response) We meant “complex combination.” Accordingly, we modified the wording. (line 28)

Introduction: line 77-79 “broad sense” is repeated twice

(Response) Thank you for the information. We rewrite the corresponding part as described above.

As a whole, the introduction should be improved as suggested above

(Response) Yes, we reorganized the introduction section.

Results: line 155: any explanation?

(Response) We added a brief summary of the discussion of the corresponding study.

Thus, quipazine acts by competing in some of the training modalities. While the mechanism remained to be elucidated, the authors discuss that single treatment with quipazine, but not quipazine+8-OH-DPAT, might interfere with the upregulation of BDNF and/or adenylate cyclase 1 induced by passive bike exercise [36].” (lines 162-166)

Line 216: unclear, please clarify

(Response) We modified the corresponding part as follows;

They showed that TMT with a customized device that enables replication of normal hindlimb movements promoted motor recovery.(lines 233-234)

Reviewer 2 Report

Comments and Suggestions for Authors

In the manuscript, the authors summarized the potential factors combined with physical rehabilitation, can lead to significant changes in motor recovery.

In chapter "3.3. Agents Targeting Inhibitory Factors in Scar Tissue 3.3.1." it is worthwhile to describe the structure of the scar tissue in a few sentences.

In the case of neurotrophic factors, I miss the description of the molecular pathways of each factor, at least in a few sentences for BDNF and NT-3.

A short description of the tables in the article would greatly help the reader to understand the table.

Overall, the manuscript provides a comprehensive overview of preclinical studies investigating the combinatorial effects of pharmacological treatment and rehabilitation following spinal cord injury.

Author Response

Reviewer 2

In the manuscript, the authors summarized the potential factors combined with physical rehabilitation, can lead to significant changes in motor recovery.

In chapter "3.3. Agents Targeting Inhibitory Factors in Scar Tissue 3.3.1." it is worthwhile to describe the structure of the scar tissue in a few sentences.

We agree on this comment. This explanation is important to promote the reader’s understanding. Accordingly, we added a few sentences regarding the structure of the scar tissue.

In the chronically injured spinal cord, glial and fibrotic scars hinder the recovery process. A persistent low-grade inflammation after subacute SCI induces both fibrotic scar remodeling by fibroblast-like cells or macrophages to express inhibitory factors to plastic changes and glial scar replacement with components such as chondroitin sulfate proteoglycans (CSPGs) which form a barrier-like structure to obstruct axonal regeneration.” (lines 268-272)

In the case of neurotrophic factors, I miss the description of the molecular pathways of each factor, at least in a few sentences for BDNF and NT-3.

We again truly appreciate this comment. We added a few sentences to explain BDNF and NT3 in brief.

BDNF acts to promote axonal growth and sprouting, as well as neuroprotective effects [41].” (lines 215-216)

The effect of NT3 on the injured spinal cord is mostly similar to BDNF. However there is a discussion regarding the nociceptive system. While NT3 ameliorates pain-related behavior, some researchers have reported that BDNF sometimes induces sensitization, particularly in an overexpression model with adeno-associated virus (AAV) [46]. ” (lines 227-231)

A short description of the tables in the article would greatly help the reader to understand the table.

We detected twelve studies in this category, and briefly summarized the basic information, the content of rehabilitation and medical treatments, and the chief mechanisms (Table 1).” (lines 120-122)

We picked up six studies investigated combinatorial effect of neurotrophic factors and rehabilitation (Table 2).” (lines 211-213)

Researchers have investigated the combinatory effect of these agents with rehabilitative interventions (Table 3). We found eight studies applying cABC to degrade CSPG, two of which use another agent targeting keratan sulfate degradation and anti-Nogo-A antibody, respectively, as a comparison. A study uses growth factor cocktail in combination with cABC as introduced above.[54] In addition, a few studies investigate unique agent.” (lines 273-278)

Many are neuroprotective, but some seem to induce neuroregenerative effects. A variety of studies in this section include: a study targeted the musculoskeletal system with testos-terone [75], a study targeted urination [76], and a series of studies utilizing induced low-grade inflammation [77][78].” (lines 375-378)

Overall, the manuscript provides a comprehensive overview of preclinical studies investigating the combinatorial effects of pharmacological treatment and rehabilitation following spinal cord injury.

We truly thank this reviewer for the time and effort to review our draft, as well as the thoughtful comments.

Reviewer 3 Report

Comments and Suggestions for Authors

This is a fairly extensive search of published literature examining pharmacological treatments with rehabilitation after spinal cord injury.  This review focused mainly on studies using a single pharmacological treatment combined with treadmill training.  The review is separated into sections, each describing a different type of pharmacological treatment and how it affected anatomical or functional recovery.   Many of the descriptions are basic with little discussion of the pros and cons of the individual studies.   The tables were helpful in filling in some of the details not in the text, such as the lesion type and location in the spinal cord.    The discussion at the mostly repeats what is in the results section of the review drawing a few general conclusions.  Much of the discussion could be incorporated into the various sections to reduce repetition of descriptions.

Line 60 – 63.  The author’s state: “Despite extraordinary efforts of clinicians and researchers to establish more effective training methods and to clarify the mechanisms underlying rehabilitation, the treatment effects are insufficient to induce remarkable functional recovery in SCI patients.”  This is not true, rehabilitation is the only know clinical treatment for spinal cord injury and in acute patients is the best known therapy for SCI patients.

Comments on the Quality of English Language

Line 155: quipazine acts by competing in some

Line 179: They showed that each of exercise, medication and

Line 267: reinforcement of one behavior may interfere with another behavior

Line271: increased the level of gross CST fibers what does this mean?  Sprouting?

Line 315: strategy closer to the clinical translation clinically relevant strategy

Line 365: number of motor neurons in the ventral horn

Author Response

Reviewer 3

This is a fairly extensive search of published literature examining pharmacological treatments with rehabilitation after spinal cord injury.  This review focused mainly on studies using a single pharmacological treatment combined with treadmill training.  The review is separated into sections, each describing a different type of pharmacological treatment and how it affected anatomical or functional recovery.   Many of the descriptions are basic with little discussion of the pros and cons of the individual studies.   The tables were helpful in filling in some of the details not in the text, such as the lesion type and location in the spinal cord.    The discussion at the mostly repeats what is in the results section of the review drawing a few general conclusions.  Much of the discussion could be incorporated into the various sections to reduce repetition of descriptions.

We sincerely appreciate the reviewer's fair evaluation of our draft. As the reviewer indicated, we initially thought that more discussion might reduce the description repetition. However, we recognized that each study's findings and main topic vary so much that the incorporation will increase the risk of misunderstanding the readers. This was why we chose to accumulate short introductions of each single study to be as precise as possible. We explained our policy shortly in the Material and Method section.

Since the topic and main result of each study vary, we chose to accumulate short intro-ductions of each single study to be as precise as possible in this review.” (lines 114-116)

Line 60 – 63.  The author’s state: “Despite extraordinary efforts of clinicians and researchers to establish more effective training methods and to clarify the mechanisms underlying rehabilitation, the treatment effects are insufficient to induce remarkable functional recovery in SCI patients.”  This is not true, rehabilitation is the only know clinical treatment for spinal cord injury and in acute patients is the best known therapy for SCI patients.

We truly thank the reviewer indicating this important point. This description intended not acute but chronic SCI. Since we noticed this was misleading to readers, we modified the corresponding part as follows;

While rehabilitation promote body functionality and activity of daily living in the acute to subacute patients, the treatment effects are insufficient to induce remarkable functional recovery in chronic phase Extraordinary efforts of clinicians and researchers to establish more effective training methods and to clarify the mechanisms underlying rehabilitation targeting chronic SCI.” (lines 63-68)

Line 155: quipazine acts by competing in some

Line 179: They showed that each of exercise, medication and

Line 267: reinforcement of one behavior may interfere with another behavior

Line 315: strategy closer to the clinical translation – clinically relevant strategy

Line 365: number of motor neurons in the ventral horn

We appreciate this reviewer for indicating grammatical errors. We correctly modified them.

Line271: increased the level of gross CST fibers – what does this mean?  Sprouting?

We apologize this non-scientific description. We modified the part referring to the original article. We consider corresponding part in the Table does not need modification.

They also found that cABC treatment significantly improved recovery of skilled paw reaching and ladder and beam walking, and both cABC treatment and rehabilitation increased the CST branching and crossing of the gray/white matter boundary and BDA-labeled axons. Furthermore, rehabilitation synergistically increased the number of CST axons at the lesion epicenter and vesicular glutamate transporter 1 expression (VGLUT1)-positive presynaptic boutons.” (lines 293-298)

Reviewer 4 Report

Comments and Suggestions for Authors

It is an interesting paper assessing a narrative review about the combinatorial effect of drugs and rehabilitation found in studies dealing with animal experiments on spinal cord injury.

The authors focus mainly on neuromodulators, neurotrophic factors and anti-glial scar agents applied in combination with rehabilitative treatments.

The title should say "Do" instead of "Does" and, although the authors exclude clinical trials in their scope, since the studies in humans are the most relevant, the authors should, at least, discuss their preclinical results with those of some distinct studies in humans.

The authors should also develop a longer section of conclusions, making it  more informative and highlighting the results after synthesizing all those reports reviewed.

Comments on the Quality of English Language

line 292: "catalize" instead of "catalizes"

line 300: Please delete the space in "functional"

pg. 12 (table 3), in the reference by 2023 Griffin: Bood-Brain-Barrier instead of   -Barieer

line 352: investigating instead of investigated

line 418: neural instead of nerval

line 554: due instead of "dues"

line 555: delete "of"

Author Response

Reviewer 4

It is an interesting paper assessing a narrative review about the combinatorial effect of drugs and rehabilitation found in studies dealing with animal experiments on spinal cord injury.

The authors focus mainly on neuromodulators, neurotrophic factors and anti-glial scar agents applied in combination with rehabilitative treatments.

The title should say "Do" instead of "Does" and, although the authors exclude clinical trials in their scope, since the studies in humans are the most relevant, the authors should, at least, discuss their preclinical results with those of some distinct studies in humans.

The authors should also develop a longer section of conclusions, making it  more informative and highlighting the results after synthesizing all those reports reviewed.

We truly appreciate the reviewer's recommendation. Indeed, it was embarrassing using “Does” instead of “Do”, we thank this reviewer informing us this mistake. Concerning the clinical trials, we initially planned to introduce some of them, though, we also discussed it may be unfair to have only a part of them because we cannot compare which pharmacological strategies are superior in truth. On this ground, we did not include human studies in this review. We are very happy if the reviewer would understand the reason. Concerning the conclusion matter, we made a figure instead of table 5 to promote the readers’ understanding. Since this review is already quite long, an illustration may be better than explanatory sentences.

In this figure, we slightly reorganized the groupings, which were previously A) Regeneration, B) Neuroprotection, C) Plasticity/Reorganization, and D) other, and modified them as 1) Regeneration of tracts, 2) Neuroprotection, 3) Scar tissue reorganization, 4) Plasticity, 5) Microenvironment in lumbar enlargement, and 6) Musculoskeletal system. In brief, previous (B) was divided into (2) and (3), previous (D) was divided into (5) and (6). Accordingly, some of the contents were changed in the group; “Neuroconductivity” was newly classified as (1), “Remodeling of the perineuronal net” as (3), and “Endogenous NS/PCs” as (4). The corresponding part in the abstract is also modified.

 “regeneration of tracts, neuroprotection, scar tissue reorganization, plasticity of spinal circuits, microenvironmental change in the spinal cord, and enforcement of the musculoskeletal system” (Abstract)

The combinatorial effects of rehabilitative training are illustrated. Beneficial effects are classified as regeneration of tracts, neuroprotection, scar tissue reorganization, plasticity of spinal circuits, microenvironmental change in the spinal cord, and enforcement of the musculoskeletal system.” (Figure 1, legend)

line 292: "catalize" instead of "catalizes"

line 300: Please delete the space in "functional"

  1. 12 (table 3), in the reference by 2023 Griffin: Bood-Brain-Barrier instead of   -Barieer

line 352: investigating instead of investigated

line 418: neural instead of nerval

line 554: due instead of "dues"

line 555: delete "of"

We truly appreciate this reviewer indicating many of our mistakes. Accordingly, we modified them all.